# INTERPRETABLE PRE-TRAINED TRANSFORMERS FOR HEART TIME-SERIES DATA

## ABSTRACT

Interpretability of artificial intelligence models is vital in healthcare, as a poorly informed decision can directly impact the health and well-being of patients. This means that, owing to their black box nature, deep-learning solutions that may even yield high accuracy often fail to be adopted in real-world healthcare settings. To this end, we employ the generative pre-trained transformer (GPT) framework to clinical heart time-series data, to create two pre-trained general purpose cardiac models, termed PPG-PT and ECG-PT. We place a special emphasis on making both such pre-trained models fully interpretable. This is achieved firstly through aggregate attention maps which show that, in order to make predictions, the model focuses on similar points in previous cardiac cycles and gradually broadens its attention in deeper layers. Next, we show that tokens with the same value, which occur at different distinct points in the electrocardiography (ECG) and photoplethysmography (PPG) cycle, form separate clusters in a high dimensional space. Such clusters are formed according to the phase of the cardiac cycle, as the tokens propagate through the transformer blocks. Finally, we highlight that individual attention heads correspond to specific physiologically relevant features, such as the dicrotic notch in PPG and the P-wave in ECG. Importantly, it is also demonstrated that these pre-trained models are straightforward to fine-tune for tasks such as the classification of atrial fibrillation (AF), and beat detection in photoplethysmography. The so introduced PPG-PT and ECG-PT models achieve accuracy comparable to the state-of-the-art for both tasks, whilst crucially retaining their interpretability and explainability. This is demonstrated in the AF-screening fine-tuned model, with attention clearly shifting to regions in the context that are strongly indicative of atrial fibrillation.

## 1 INTRODUCTION

The generative pretrained transformer (GPT) (Radford & Narasimhan, 2018) (Brown et al., 2020) is a large language model (LLM) which forms the basis of the chat-GPT models (OpenAI, 2024). It consists of layers of decoder-only transformers (Vaswani et al., 2017) that are trained to predict the next token (words, sub-words, or characters), using all previous tokens provided to the model as context. The attention mechanism is masked, such that tokens inputted to the model can only communicate with past tokens and thus a context window of length $N$ effectively provides $N$ different training examples, whereby the model tries to predict the same context window shifted by one token into the future. The whole premise of the GPT model is that in order to accurately predict the next token, across a large and diverse array of texts, a model must gain an efficient and general comprehension of these texts. Given that language is a tool that we use to summarise and communicate the world around us, this efficient and general understanding allows the pre-trained model to be fine tuned for a myriad of complex tasks (OpenAI, 2024).

The assertion of this work is that we can leverage the GPT mechanism, i.e. stacks of decoder-only transformers trained to predict the next token, and develop it for physiological time series. Our hypothesis is that if our models gain an efficient and general understanding of a physiological time series, such as photoplethysmography (PPG) or electrocardiography (ECG), then these models will be simple to fine-tune for downstream tasks such as the classification of heart conditions. The ECG signal refers to the monitoring of the electrical activity of the heart through external non-invasive electrodes that measure the potential difference across the heart (Arora & Mishra, 2021; Davies

et al., 2024a). In this work, we use only single-lead ECG. The photoplethysmography (PPG) signal refers to the non-invasive measurement of blood volume (Charlton et al., 2023). The PPG operates by shining light through tissue with a light emitting diode, which is absorbed by the blood, and then either measuring the transmitted light or reflected light with a corresponding photo-diode. The PPG is thus able to measure changes in blood volume related to pulse and even breathing (Davies et al., 2022). In PPG, for example, in order to accurately predict the next time point the model must gain knowledge on several underlying phenomena, such as the subtle changes in heart rate and pulse amplitude due to breathing and blood pressure. This leads us to postulate that a large model that is good at predicting the next token of PPG will be straightforward to fine-tune for specific PPG health-related tasks.

As GPT and other large language models are rapidly gaining in capability, and thus decision making power, it is of crucial importance that LLMs do not remain black boxes. Indeed, the black box nature of deep learning models is an immediate problem in healthcare, where decisions made by artificial intelligence (AI) have the potential to directly impact our health and even our lives. It is therefore paramount that models are developed which give clear insights and explanations into why decisions are made. A natural way to interpret the operation of transformer-based LLMs is to take a close examination of the attention weights of the individual transformer blocks. This attention mechanism provides information on which tokens a model looks at in order to make a decision, and thus it is common to use heat maps that visualise this attention as an avenue for interpretability (Zhao et al., 2023). This has been applied to great effect in large vision language models (LVLMs) (Stan et al., 2024) where it is possible to visualise the focus points on an image based on the context of a language prompt. Language is complex, and there are often several pathways to achieve the same result. In many instances this makes it difficult to decipher attention, which means that in the context of LLMs attention cannot always be relied upon for explainability (Jain & Wallace, 2019). However, like images, both the PPG and ECG signals are far less complicated than language, and downstream tasks, such as screening for heart conditions, offer well understood constraints on the possible pathways to achieve a correct diagnosis. This claim is evaluated in Appendix A.1, where it is demonstrated that the possibilities for the next token of PPG become highly constrained as context length increases. It is therefore a core focus of this paper not just to provide generalised transformer models for PPG and ECG related tasks, but to provide models that are fully interpretable so that they may be safely used in a healthcare setting.

To this end, we firstly develop two pre-trained decoder-only time series transformer models for use with heart signals, namely PPG-PT and ECG-PT. Furthermore, we demonstrate that both the PPG-PT and ECG-PT models behave exactly as expected for the task of predicting the next time series point in PPG and ECG through:

- Aggregate attention of different transformer layers.
- Changes in cosine similarity between core PPG and ECG features in the embedding space upon propagation through transformer layers.
- The analysis of the attention weights of individual attention heads in the final transformer block.

Finally, we demonstrate how these models can be fine-tuned to detect atrial fibrillation (AF), a common type of abnormal heart rhythm. The changes in attention from the task of next token prediction to the task of classification of AF can be used to further explain the reason why a model has made a decision. This makes it possible to provide clinicians with both accurate classification and the reasoning behind it, thus allowing for a conjoint operation between a clinician and AI in diagnostics and treatment through informed decisions.

## 2 RELATED WORK

There are several valuable prior works which leverage self supervised learning (SSL) to train large models for application to ECG and PPG (Abbaspourazad et al., 2024; Cheng et al., 2020; Spathis et al., 2021). Pre-trained transformer architectures have previously been shown to be effective for PPG (Chen et al., 2024), but tokenise based on patches rather than individual time points, thus limiting model interpretability. Another such pre-training method is contrastive learning (Kiyasseh et al., 2021), shown to be effective for pre-training a generalisable representation of ECG signals.

Whilst these models provide impressive accuracies on a wide range of downstream tasks, none of these works were designed for end user interpretability. Where attempts at interpretability exist, it is through class activation maps (CAM) to visualise regions in the input time-series that are of high importance to classification (Lai et al., 2023; Torres-Soto & Ashley, 2020). However, the regions of high intensity in the generated interpretability heat-maps are either too broad or have peaks at regions that are not of interest (Lai et al., 2023). Moreover, the aforementioned works either do not use publicly available datasets, or provide model files or model interpretability code. This highlights a clear need for models pre-trained on bio-signals that are transparent and interpretable. With this in mind, the models presented in this work are designed with interpretability and explainability as the primary goal. For generality and reproducibility, we use publicly available datasets, provide all code and trained model files and even graphical user interfaces explicitly created for interpretability.

## 3 THE PRE-TRAINED TRANSFORMER MODELS

### 3.1 TOKENISATION AND EMBEDDING

Both the ECG and PPG signals are largely locally periodic, and in most applications we can ignore any mean offset. This allows us to divide each PPG and ECG signal into a range of finite tokens, forming a vocabulary from which each signal can be constructed. To tokenise the PPG signals, they were resampled to 50Hz and each 10-second context window (500 samples) was scaled to between 100 and 0. The resulting signal was then rounded to integer values, yielding a total vocabulary size of 101 tokens. Importantly, in our models each sample corresponds to a token. The 10-second context window was chosen so as to be long enough to preserve low frequency variations in the PPG (such as respiratory variations), but not too long so as to run into memory issues when training. For the ECG, the process was similar, apart from resampling to 100Hz and thus using a context window that corresponded to 5-seconds instead of 10-seconds of data. A higher sampling frequency was required in the ECG tokenisation in order to preserve the high frequency components of the QRS complex. In GPT models, common sequences of token values can be combined during tokenisation, in order to allow for a longer context length for the same number of tokens. This method of tokenisation was not implemented in the PPG-PT and ECG-PT models, as further extending the length of the context window was not necessary.

Upon tokenisation, each token was embedded with a token embedding table and a position embedding table. The dimensions of the token embedding were the vocabulary size times the embedding dimension ($d_{model}$), giving a $d_{model}$-dimensional vector for every possible token in the vocabulary. The dimensions of the position embedding table were the maximum context length times $d_{model}$, giving a $d_{model}$-dimensional vector for every possible position in the context window. In the original transformer paper, the positional embeddings utilised sine and cosine functions of different frequencies (Vaswani et al., 2017); whereas in our models, both of these embeddings are learnt as the model trains. The token and positional embeddings were added together, which means that the attention mechanism in the subsequent transformer blocks had information on both the specific token and the position of the token in the context.

### 3.2 ARCHITECTURE AND TRAINING

Our models were developed in PyTorch (Paszke et al., 2019) and adapted from a tutorial by Andrej Karpathy (Karpathy, 2023) titled "Let's build GPT: from scratch, in code, spelled out". For both the photoplethysmography pre-trained transformer (PPG-PT) and electrocardiography pre-trained transformer (ECG-PT) we used an embedding dimension ($d_{model}$) of 64. The original GPT paper used an embedding dimension of 768 (Radford & Narasimhan, 2018), but given that PPG and ECG are far less complex than language, $d_{model} = 64$ was found to be sufficient in our case. For our decoder-only transformer we used 8 transformer blocks, each with 8 attention heads for the multi-head attention as described in the original transformer paper (Vaswani et al., 2017) and in Appendix A.2. In the areas where dropout was applied (the feedforward subsection of the transformer and the attention weight matrix), the dropout was set to 0.2. Our models use a context length of $N = 500$ samples, corresponding to 5-seconds of ECG, resulting in 443,493 trainable parameters.

For pre-training, multiple publicly available datasets from different regions of the world were combined for both the ECG and PPG models, resulting in 128 million tokens for PPG-PT training and 42

million tokens for ECG-PT training and corresponding to a similar ratio between the number of parameters and data-size as state-of-the-art LLMs (Dubey et al., 2024). The full pre-training details are given in Appendix A.3. Importantly, unlike conventional time-series transformer models (Das et al., 2024), we opted to train our models purely with cross-entropy loss and next token prediction, rather than mean squared error (MSE) and windowed prediction (see Appendix A.4). This, combined with the locally periodic and deterministic nature of PPG and ECG, resulted in the interpretable attention mechanism that we demonstrate in this work.

For fine-tuning, for both tasks of screening for atrial fibrillation and beat detection in PPG, only the last layer of the pre-trained model was fine-tuned. In addition, the final linear layer which transforms the model dimension into the vocabulary size was replaced my a linear layer which transforms the model dimension into 1-dimension. The full fine-tuning methodology, including details of the publicly available datasets used, are provided in Appendix A.5.

## 4 GENERATIVE PRE-TRAINED MODEL EVALUATION

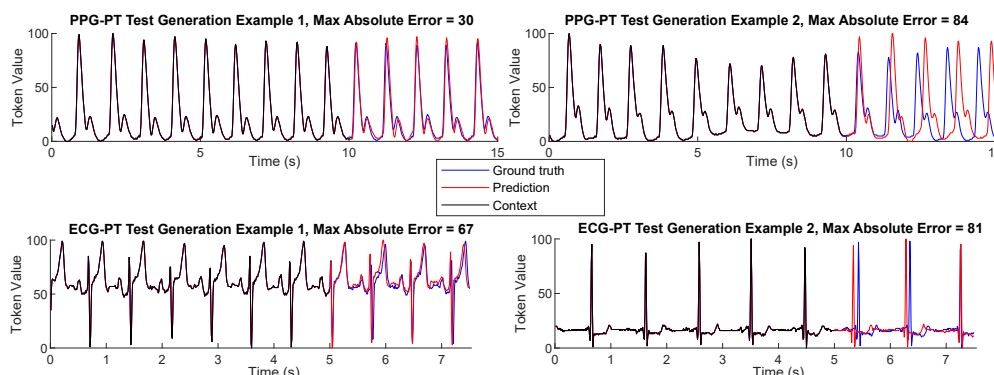

Figure 1: Example generations and the corresponding maximum absolute errors for both the PPG-PT and ECG-PT base models for a duration of half the length of the context window. The examples demonstrate that whilst the models were able to accurately capture features of the context, such as pulse shape and the dicrotic notch in the case of PPG, and all sections of the P-QRS-T complex for the ECG, large errors can arise from slight temporal misalignment. Two such examples of model generations for both PPG-PT and ECG-PT are provided with context in black, ground truth in blue and model prediction in red.

In training, for all points in the context window the model outputs logits which are transformed into a next token probability distribution. For the generation of future tokens, the probability distribution corresponding to the last point in the context is taken. The `multinomial` function in PyTorch is then used to sample this distribution and generate a predicted next token. The single generated token can then be appended to the previous context, which is then used to generate another token. This process can be repeated, generating one token at a time, until a maximum number of generated tokens has been reached. Another way to examine the predictive capabilities of each model would be to predict the next token as the one with the highest probability, instead of sampling from a distribution. The problem with this second approach is that sometimes in low frequency periods, such as a trough in PPG or a period between P-QRS-T regions in the ECG, taking just the token with maximum likelihood could allow the model to become stuck in predicting the same token value over and over again. Because of this issue, the first approach of sampling from a probability distribution was chosen to evaluate the models.

The full dataset and pre-processing details for the evaluation of our pre-trained generative models are provided in Appendix A.3.1. For each predicted token, we calculated the absolute distance between the predicted token and the true token. We were then able to calculate the median and inter-quartile range of the absolute prediction error against the prediction horizon for both the PPG-PT and ECG-PT models. In Figure 1, example generation waveforms and maximum absolute errors are shown for both the PPG-PT and ECG-PT models. Observe that the errors stem not from a failure to predict the morphology of the PPG or ECG, but from slight temporal misalignment between the prediction

and the ground truth, as is common with periodic physiological signals (Davies et al., 2024b). The median absolute distance between the prediction and ground truth in PPG-PT starts at 1 token and grows to 21 tokens across the 5-second prediction horizon (250 samples). For ECG-PT, the median error starts at 1 token and grows to 3 tokens across the 2.5-second prediction horizon (250 samples). This aggregate analysis is provided in Appendix A.6. It should be noted that we expect the errors to be significantly higher in the PPG-PT model when compared with the ECG-PT model, as due to the comparatively broader peaks in the PPG waveform, errors in temporal misalignment increase the prediction error across the whole waveform. This is in contrast to ECG where the vast majority of the signal is at baseline, and thus slight temporal misalignment is mainly penalised around the QRS complex and the T wave.

Whilst the prediction errors in Figure 1 show us that the introduced models are good at extrapolating PPG and ECG waveforms into the future, this can never be perfect given that there may be changes in heart rate, respiration, and even arrhythmia events that are, by definition, impossible to predict from the context. The important result here is the ability of the models to generalise to unseen distinct morphologies of PPG and ECG as context. Observe from Figure 1 that the PPG-PT condenses knowledge of peak width and the shape of the dicrotic notch (the sub-peak) in the PPG examples and is thus able to generate PPG with the same properties. This is even more obvious in the case of ECG, where both examples have very different P, Q, R, S and T wave morphologies, and ECG-PT was able to replicate this well. This is the first indicator that the PPG-PT and ECG-PT models indeed pay attention to the relevent features of both signals.

## 5 INTERPRETABILITY OF THE PRE-TRAINED MODELS

### 5.1 INTERPRETABILITY OF AGGREGATE MODEL ATTENTION

As is the case with GPT models (Radford & Narasimhan, 2018), our pre-trained transformer models (PPG-PT and ECG-PT) are trained to predict the next token at each point, by using the attention mechanism to collate knowledge from previous tokens. Given the task of predicting the next token in a periodic signal, it is natural to think that a model or a human would look at tokens which are at the same point in the previous cardiac cycles in order to make that prediction. For example, if the current token to predict was a peak, it would be logical to look at all previous peaks, with an emphasis on peaks of similar height and width, in order to make an accurate prediction. However, the information we start with is just the position of a token in the context window and the value of a token. Given that the fundamental frequency changes between contexts and that different points in the local cycle can have the same value, we cannot rely on the initial position and value information alone to find similar points in the same cycle. In order to understand the point at which a token lies in a cardiac cycle, it is therefore necessary to first look at the context of surrounding tokens. Once this relationship between a token and its local cycle is understood, attention can broaden to look at all cardiac cycles.

This mechanism of broadening attention is indeed found in both of our PPG-PT and ECG-PT models. The last row in the attention weight matrix corresponds to the final token in the context. To examine the attention span of the model, we calculated the central point of attention for this final row of attention in each attention head in each transformer block, across all context windows in the generative test set. Table 1 (Appendix A.7) shows that for PPG-PT, the mean central point of attention is 0.43 seconds in the first transformer block, and thus the focus is within the immediate cycle. By the last transformer block, this central point of attention has broadened to 2.31 seconds, indicating that the PPG-PT model was updating the current token based on the tokens in previous cycles of PPG. The same effect is seen in ECG-PT, with a broadening of attention from 0.33 seconds in the first block to 1.88 seconds in the last block (the typical duration of a cardiac cycle is around 0.8-1 seconds).

The next step to interpreting the aggregate attention of the model is to examine the attention weights of different transformer blocks for specific context examples. In Figure 2 (a) we examine the summed attention weights in PPG-PT from the row associated with predicting the peak point highlighted with a blue circle. As expected, in order to predict the peak point the first transformer block focuses on the immediate points before the peak point in order to learn how the point integrates into the local PPG cycle. This also is true for the attention matrix rows in the first transformer block that are associated with all previous tokens in the context. Importantly, observe that in the final

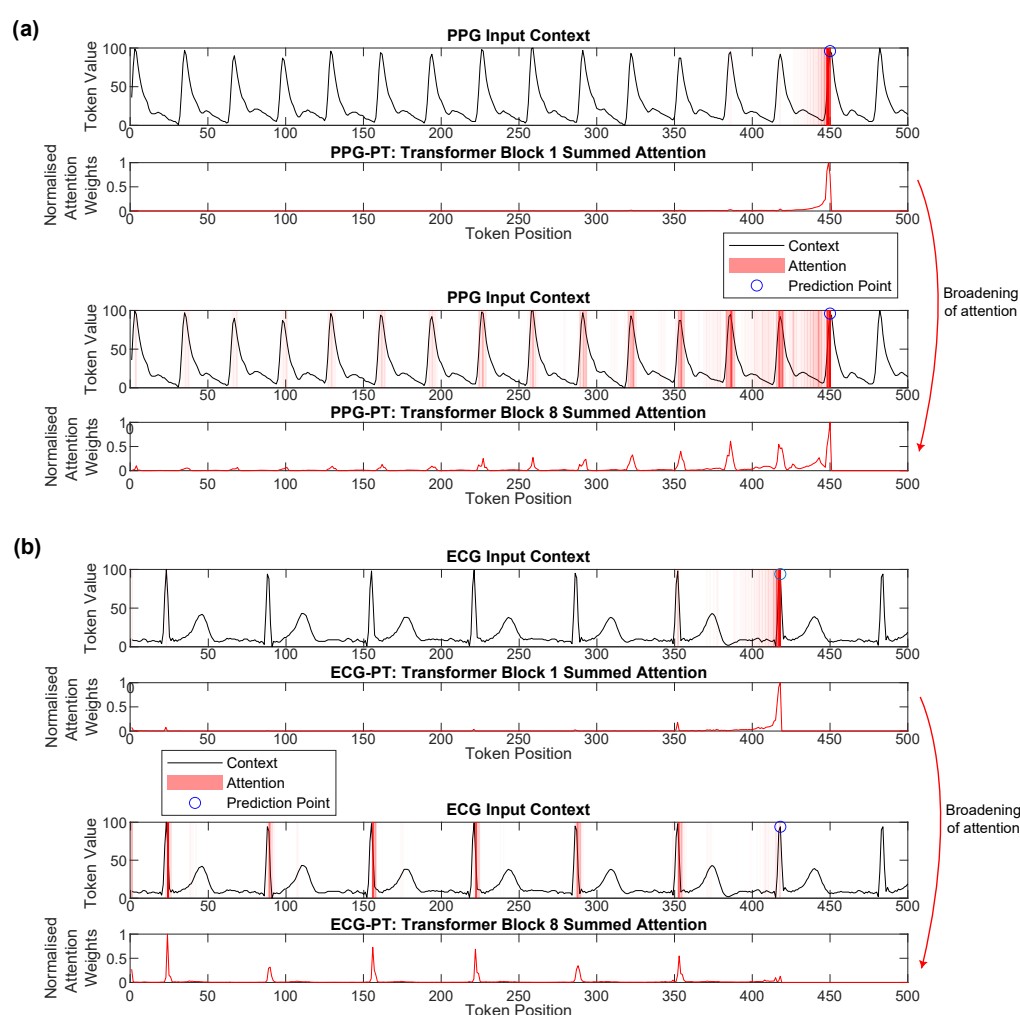

Figure 2: Aggregate attention for both the PPG-PT and ECG-PT models across all attention heads for the attention row corresponding to the prediction point, per transformer layer shown (the first and last layers). The attention maps demonstrate that in order to predict the next token, the models first look at all tokens in the local cycle to gain an understanding of where a token falls within that cycle. When this understanding has been established, models then look at the similar points occurring in other cycles in the context window. For both models, attention in the first transformer layer is shown on top and the last transformer layer beneath, with context (black line) and the prediction point (blue circle) corresponding to overlaid transformer attention (red transparent bars), with transparency scaled based on the attention weights shown below (red solid line). (a) The PPG-PT aggregate attention, for the prediction of a peak in the given previously unseen photoplethysmography context. (b) The ECG-PT aggregate attention, for the prediction of a peak in the given previously unseen electrocardiography context.

transformer block, in order to predict the peak point the network focuses its attention on all previous peaks in the context. This is also true for the ECG-PT model, as demonstrated in Figure 2 (b).

## 5.2 VECTOR SIMILARITIES BETWEEN POINTS OF INTEREST

In the previous section, we have demonstrated that the pre-trained transformer models naturally focus on similar points in previous cycles of PPG or ECG when predicting the next token. This strongly indicates that the models are able to distinguish between different points in the cycle of PPG and ECG waveforms. In this section, we aim to solidify this finding by examining how the

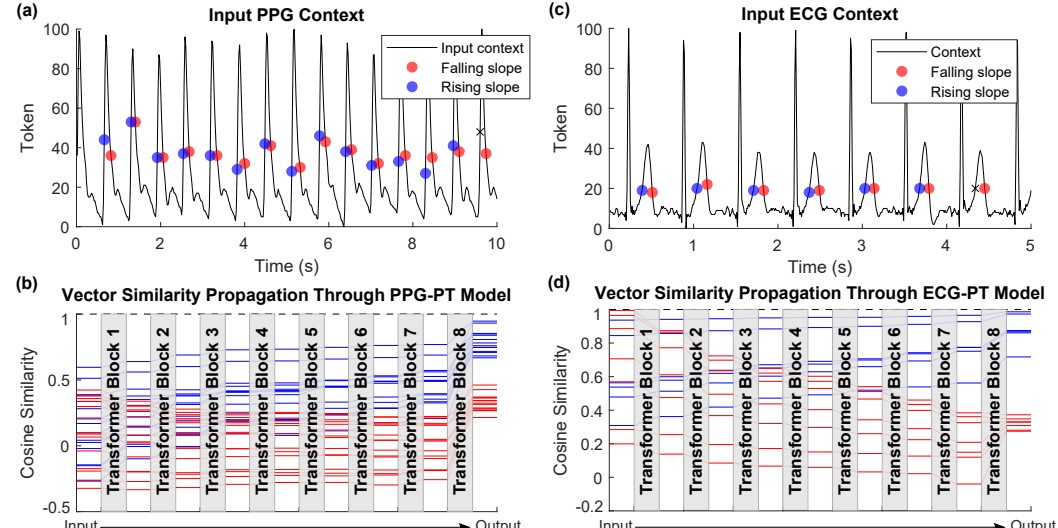

Figure 3: Cosine similarity of points in rising slopes (blue circles, blue solid lines) and falling slopes (red circles, red solid lines) with similar input values, upon propagation through the transformer layers of the model. In each case, a single point on a rising slope is chosen as a comparison point, represented by a black cross on the context (black solid line), and a black dotted line with a cosine similarity of 1 in the cosine similarity plot. The point of propagation through the model is highlighted with grey transparent blocks labeled with the corresponding transformer layer (from 1 to 8). In both examples, rising slope points and falling slope points are shuffled in the embedding space upon input to the first model layer, and gradually divide into two clusters as they propagate through the model. (a-b) The cosine similarity of rising slopes and falling slopes in a photoplethysmography signal, upon propagation through the PPG-PT model. (c-d) The cosine similarity of rising slopes and falling slopes on the T-wave of an electrocardiography signal, upon propagation through the ECG-PT model.

cosine similarity of embedded tokens with the same value, which occur at different distinct points in the PPG and ECG cycle, change upon propagation through the model. To this end, we chose tokens at similar points on rising slopes and falling slopes on the PPG signal, and on the T-wave of the ECG signal. The similarity of these tokens was calculated with respect to the final rising slope in each context window. If the models are effective at distinguishing different points in the cycle, we would expect rising slopes to cluster together in high-dimensional space, and falling slopes to form a separate cluster.

**Remark 1.** *An analogous experiment in a large language model would be to look at the vector similarity of homonyms (words that have the same spelling but can have multiple different meanings), and examine how the vector similarity changes based on specific context in a sentence. For example, we would expect the token "bat" to form separate clusters upon propagation through an LLM based on if it contextually refers to a "club to hit a cricket ball" or it refers to the "flying mammal".*

It is highlighted in Figure 3 that, upon input to the pre-trained models, falling slopes and rising slopes of the PPG and ECG are shuffled in high-dimensional space. This is because the inputs to the models have the same token embedding that has not yet been updated based on the context of previous tokens. Notably, as these tokens propagate through the models, rising slopes increase in vector similarity and cluster below a cosine similarity of 1 (identical vectors) and falling slopes form their own independent cluster. This further demonstrates that these large pre-trained models can clearly identify the relationship between different points in cycle of both PPG and ECG.

## 5.3 ATTENTION MAPS OF INDIVIDUAL ATTENTION HEADS

Now that we have established that the pre-trained PPG-PT and ECG-PT models attend to similar points in previous cycles in order to predict the next token, and that the models have knowledge of the

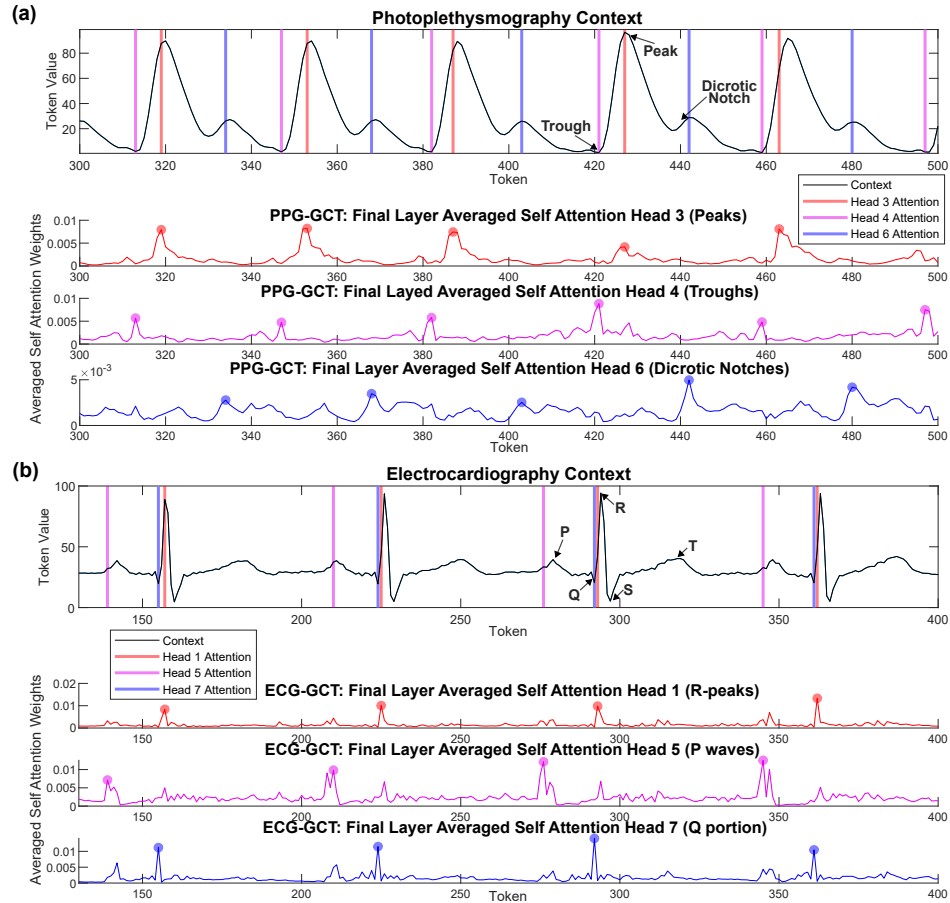

Figure 4: Attention maps of the attention weights of individual attention heads of the last layer of both the PPG-PT and ECG-PT models. Averaged attention maps are generated by averaging the final layer of the attention weight matrix over the prediction of the next 500 tokens beyond the initial context window. The model context is plotted in black, with attention weights plotted in red, pink, and blue for different attention heads. Peaks in attention are highlighted in the same colour with circles, and overlaid on the context with bars. (a) Mapping of the averaged attention weights in the final layer attention heads of the PPG-PT model for an example photoplethysmography context. Observe that the 3rd attention head (red) looks primarily for peaks in PPG, the 4th attention head (pink) looks primarily for troughs in PPG, and the 6th (blue) looks for the dicrotic notch. (b) Mapping of the averaged attention weights in the final layer attention heads of the ECG-PT model for an example electrocardiography context. The 1st attention head (red) looks for R peaks in the ECG, the 5th attention head (pink) looks primarily for P waves, and the 7th attention head (blue) looks primarily for the Q portion of the QRS complex of ECG.

context of different points within a PPG and ECG cycle, the final step is to examine if the individual attention heads attend to specific high level features in the PPG and ECG signals. To investigate this, we examined example contexts from the generative evaluation dataset which contained the common features of each signal. For example, a PPG context which has well-defined peaks, troughs, and a dicrotic notch, and an ECG context which has a well defined P-wave and QRS complex.

For each model, we averaged the final row of the attention weights by shifting our context in time, to build up an aggregate map of which regions of the context an individual head responded to. This method is summarised in Appendix A.14. Regions of interest were highlighted by implementing the `findpeaks` function in MATLAB, on the averaged attention map, with a minimum peak distance of 15 tokens and a minimum peak height of 0.5 of the maximum attention weight in the window, to extract the local maximas in attention.

Figure 4 demonstrates that individual attention heads do indeed pay attention to important features in the signal of interest. In particular, PPG-PT has an attention head that looks for peaks, another that looks for troughs and another which looks for the dicrotic notch. In addition, we found that ECG-PT has a head which looks for the Q portion of the QRS, another which looks for R-peaks and another that looks for P waves. Therefore, in addition to an ability to distinguish all separate points in a PPG and ECG cycle, the pre-trained models presented in this paper also pay specific attention to some of the most physiologically relevent features.

# 6 FINE-TUNING FOR AUTOMATIC SCREENING OF ATRIAL FIBRILLATION

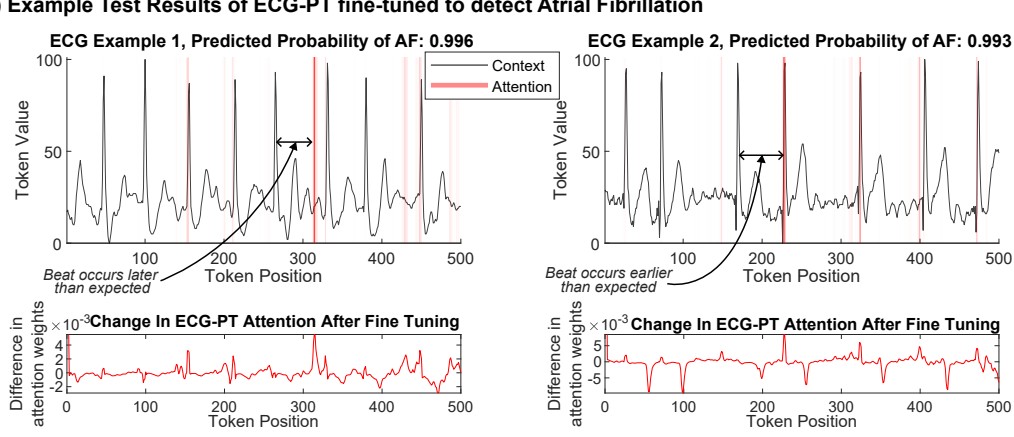

Figure 5: Changes in final layer attention weights of the base PPG-PT and ECG-PT models when fine-tuned to classifying atrial fibrillation. The model input context is shown in black, with attention overlaid with red bars of varying transparency based on the change in attention weights. The full difference in attention weights is shown below each plot in red. (a) Example fine-tuned PPG-PT test results, with examples of pulses occurring later than expected and earlier than expected, and the corresponding spikes in attention weights. (b) Example fine-tuned ECG-PT test results, again showing beats occurring later or earlier than expected with the aforementioned spikes in attention.

**Remark 2.** *Through pre-training we have obtained models which focus on the important features of PPG and ECG, and generalise well to unseen morphologies. However, rather like in LLMs, predicting the next token of PPG and ECG lacks utility. To make full use of these pre-trained models, we fine-tune the models to classify atrial fibrillation (AF), as outlined in Appendix A.5.*

Atrial fibrillation is the most common arrhythmia, characterised by an irregular heart rate which manifests itself in rapid increases in heart rate and periods where it slows down dramatically (Wijesurendra & Casadei, 2019). The fine-tuned AF-PPG-PT model achieved a leave-one-subject-out

area under the curve (AUC) of 0.93 when cross-validated across all test subjects, and 0.96 when two subjects with poor signal to noise ratio were excluded. The fine-tuned AF-ECG-PT model achieved an AUC of 0.99 when cross-validated across all test subjects. A comparison between our results and the state-of-the-art is provided in Appendix A.8. Further fine-tuning experiments, for beat detection (Appendix A.9), for ventricular premature contractions along with corresponding interpretability analysis (Appendix A.10) and for four class arrhythmia classification (summarised in Appendix A.11). It is worth reiterating that these results were achieved with 11 minutes of fine-tuning computer time, corresponding to roughly 0.2% of the total time it took to train the base models.

The key result of fine-tuning is not just the accuracy, but the interpretability of the results. To ascertain why a model made a classification in the way it did, we simply aggregated the final row of SoftMaxed attention weights across all heads in the final transformer layer for the base model and compared with the fine-tuned model. Any increases in attention weights therefore indicate that the model deems the corresponding tokens valuable for classification of AF. In Figure 5(a), two representative attention maps are displayed for the classification of a PPG context as AF. Observe in both cases that the model shifts attention to periods where beats occur later than expected and earlier than expected, which is the obvious characteristic of AF. This is even clearer in the fine-tuned ECG attention, given that peaks in ECG are more precisely localised in time. In Figure 5(b), an example is highlighted where a beat is expected to occur based on the previous beat timing, but it does not and thus the model attention spikes. A second example is also highlighted, where based on the previous beat timing a beat occurs much earlier than expected, and model attention therefore spikes exactly at this point. These interpretable maps of shifts in attention, which demonstrate the reason why the model has made the classification, can easily be provided along with the probability of AF.

## 7  CONCLUSION

This work has demonstrated that GPT-like models, when trained to predict the next token locally in periodic physiological time series such as PPG and ECG, can be fully interpretable in their operation. This has been illustrated through aggregate attention maps, which are natural for the task of predicting the next token, and through the clustering of different PPG and ECG features in high-dimensional space. This has been further demonstrated through individual attention heads that correspond strongly to specific features, such as the dicrotic notch in PPG or the P-wave in ECG. Moreover, we have shown that the so introduced interpretability is carried forward when fine-tuning for abnormal heart rhythms. Indeed, in the classification of atrial fibrillation, attention shifts to regions in the input context that most indicate the presence of the arrhythmia. This work represents a step forward in the interpretability and explainability of large transformer networks when applied to healthcare settings.

### REPRODUCIBILITY STATEMENT

The code is provided for pre-training and fine-tuning, in addition to all of the pre-training and fine-tuned PyTorch model files. All datasets used are publicly available. Furthermore, we have developed five Python-based graphical user interfaces (GUIs) to allow researchers to more easily implement and interpret the models: i) A GUI which allows users to load the pre-trained models and generate next tokens based on inputted context of either PPG or ECG; ii) A GUI which allows users to visualise the attention weights of the pre-trained models for a context of their choice; iii) A GUI which allows users to visualise how the cosine similarity changes between different tokens as they propagate through the pre-trained models; iv) A GUI for the fine-tuned atrial fibrillation models, providing a probability of atrial fibrillation along with a visualisation of how final layer attention weights change from next token prediction to screening for atrial fibrillation; v) A GUI for beat-detection in PPG, allowing users to load in long sequences of finger-PPG for automatic estimation of beats and signal quality. The code, model files and GUIs will be provided in the form of a publicly available Github repository.

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

## A APPENDIX

### A.1 ESTIMATING THE ENTROPY OF PHOTOPLETHYSMOGRAPHY

The assertion is made in this work that locally periodic signals such as photoplethysmography are easier to extrapolate than language. In language there are often several valid options for the next token, with an estimate for the lowest achievable cross-entropy loss of 1.69 (Hoffmann et al., 2022), corresponding to roughly 1 in 5 predicted tokens being correct. In our pre-training, we achieved a validation loss of 1.2 for PPG (1 in 3 tokens correct) and 1.8 for ECG (1 in 6 tokens correct), with a comparatively small model (under half a million parameters). In a further attempt to validate our assertion that such signals are easier to predict than language, we analysed the first 20 million tokens of training data for PPG-PT. The data was firstly rounded into 10 possible token "bins" from the original 101 possible tokens, in order to allow for an increased likelihood of identical sequences. The frequency at which sequences occurred in the dataset were counted, and the 500 most frequently occurring sequences were stored along with the next token bin in every example the sequence was found. For every sequence that occurred more than 10 times in the training set, the probability distribution for the next token bin was calculated and then ranked from highest to lowest probability. This ranked distribution was then averaged over all sequences, and this analysis was repeated for

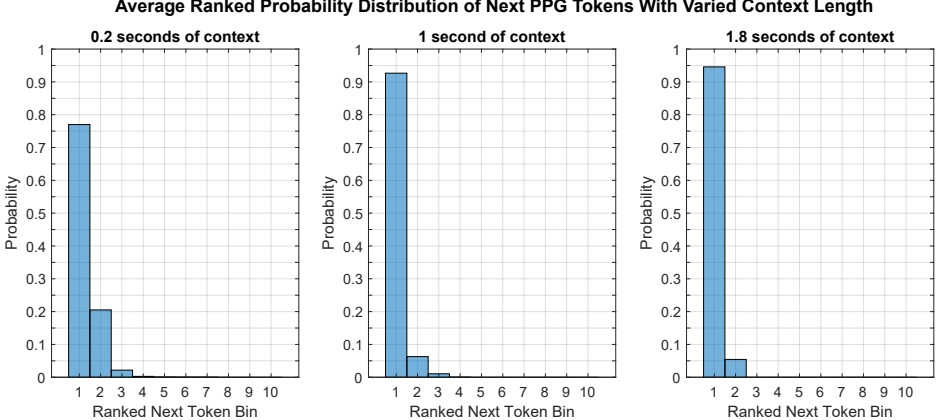

Figure 6: Estimation of the predictability of the photoplethysmography signal, rounded to 10 possible tokens, for different context lengths. Each plot shows averaged ranked probabilities of next tokens, for all sequences which were repeated up to 10 or more times (up to a maximum of 500 unique sequences) in the first 20 million tokens of training data. Left) Average ranked next token probabilities given 0.2 seconds of context. Middle) Average ranked next token probabilities given 1 second of context. Right) Average ranked next token probabilities given 1.8 seconds of context.

three different context lengths of 10 samples (0.2 seconds), 50 samples (1 second) and 90 samples (1.8 seconds).

The results of this analysis are presented in Figure 6, which shows that with a context length of 0.2 seconds, if one were to always predict the next token bin with maximal likelihood for a given sequence, they would be correct 77% of the time. This increases to 93% with 1 second of context and 95% with 1.8 seconds of context. Vast increases in predictability were found around the 1-second of context, which captures at least one cycle of PPG in the majority of cases and thus provides the knowledge of position in the cardiac cycle. It is plausible that a further jump in accuracy could occur with knowledge of the position in the respiratory cycle at context lengths beyond 5-seconds, but in our case the total volume of data was too small to find a sufficient number of repeated sequences for this analysis at long context lengths. The analysis demonstrates the highly deterministic nature of photoplethysmography, with a well-defined correct next token given a long enough context.

### A.2 MULTI-HEAD MASKED SELF ATTENTION AND THE TRANSFORMER BLOCK

Once tokens have been embedded with a token embedding and positional embedding which are added together, this allows the model to use information on both the position of the token in the context window (length $N$) and the value of the token; these are fed to transformer blocks in their high-dimensional vector form. In multi-head attention, the embedding space (of size $d_{model}$) is divided into lower-dimensional spaces of equal size ($d_k$), before the results of attention are concatenated after the attention mechanism to reconstruct the original embedding space size ($d_{model}$) (Vaswani et al., 2017).

The attention mechanism operates by allowing tokens to communicate with one another and thus update each other's information. Each attention head transforms tokens into keys ($K$), using a linear layer which compresses the tokens from $d_{model}$ dimensions to $d_k$ dimensions. It also separately transforms tokens into queries ($Q$), which again use another linear layer to compress the tokens from $d_{model}$ dimensions to $d_k$ dimensions. Queries ($Q$) can be thought of as each token broadcasting what it is looking for, and keys ($K$) as each token broadcasting what it contains. The dot product is then taken between both the queries and the keys, such that if there is a match between the information a token is looking for in $Q$ and the information a token emits in $K$, there will be a spike in the corresponding attention weights $QK^T$. This dot product is next scaled by $\sqrt{d_k}$, and the SoftMax of each row is taken (resulting in attention rows that sum to 1). This scaled dot product results in an $N \times N$ matrix of attention weights, which conveys how tokens communicate with each other,

for all tokens in the context window. In masked self attention, the mechanism used to train GPT models, the upper triangle of this weight matrix is set to zero. This means that tokens can only communicate with themselves or with past tokens in the sequence, and allows for efficient training by allowing for N separate training examples with an input block of N token length. During training of the model, dropout (the random zeroing of weights) is also applied to the weight matrix. The final step in the masked self-attention process is to multiply the weight matrix with a value matrix. The value matrix, $V$, is formed from another linear layer that compresses the tokens from $d_{model}$ dimensions to $d_k$ dimensions. A combination of all of these steps produces the following attention function (Vaswani et al., 2017)

$$\text{Attention}(Q, K, V) = \text{SoftMax}(\frac{QK^T}{\sqrt{d_k}})V \quad (1)$$

As previously mentioned, the results of each head, which are of dimension $d_k = d_{model}/(\# \text{ Heads})$, are next concatenated to give the original dimension $d_{model}$. This output from multi-head attention is then added to the input to the multi-head attention, forming a residual connection that allows the model to bypass a transformer block if needed. Finally, this new combined output is normalised and passed through a feed-forward network which firstly expands the embedding dimension to $4 \times d_{model}$, applies a ReLU activation function, and then compresses the dimension back to $d_{model}$. During training, dropout is also applied to this feed-forward network. The feed-forward network after multi-head attention can be thought of as allowing the model to process the results of self-attention, before these results are fed into another multi-head attention block. The combination of the multi-head attention and this feed-forward network constitute one transformer block. After the final transformer block, the output is passed through a final linear layer, which converts the output from a dimension of $d_{model}$ to the size of the vocabulary. When passed through a SoftMax activation function, this provides the probabilities for all tokens in the vocabulary, at all $N$ points.

### A.3 PRE-TRAINING DATASETS AND METHODS

Three datasets were used for training the PPG-PT base model: 1) Capnobase "respiratory benchmark" dataset (Karlen et al., 2013), which consists of high quality ECG and PPG recorded over 42 subjects for 8 minutes each; 2) BIDMC[1] dataset (Pimentel et al., 2016) which consists of PPG and ECG from 53 subjects for 8 minutes each; 3) the "cuffless blood pressure dataset" (Kachuee et al., 2015), a subset of MIMIC II (Goldberger et al., 2000), consisting of 12,000 recordings of PPG signals of varied quality, in a hospital setting. The combination of all of these datasets resulted in over 128 million total tokens for training.

For the training of the ECG-PT base model, we used subsets of the "PhysioNet Computing in Cardiology Challenge 2020" dataset (Alday et al., 2020; Goldberger et al., 2000), which consists of tens of thousands of 12-lead ECG recordings, across tens of thousands of patients in a hospital setting. From each of these recordings, we extracted 10-second examples of Lead I ECG. This dataset comprises a diverse range of cardiac abnormalities as well as many healthy subjects, as the dataset was originally constructed for the identification of different heart conditions. Once tokenised, this dataset resulted in over 42 million tokens for training.

For each model, training data was split into 90% training and 10% validation datasets. The data was not shuffled to ensure that validation data primarily consisted of unseen subjects. The PPG-PT model was trained over 500,000 iterations with a batch size of 64, and the ECG-PT model was trained over 560,000 iterations with the same batch size. After every 2,000 iterations, the models were evaluated over a further 200 iterations to measure validation loss. In both cases, the learning rate was set as $3 \times 10^{-4}$ within the AdamW optimiser. The training of PPG-PT took just over 5 days on an RTX A2000 12GB GPU, while the training of ECG-PT took almost 6 days. Optimisation loss was measured using cross-entropy, by mapping the $N \times 101$ model outputs (logits) where the 101 represents all possible tokens, to the target tokens which were the next token values for all points in the input. The final validation loss of the PPG-PT model was 1.2, corresponding to roughly a third of prediction being perfect. The final validation loss of the ECG-PT model was 1.8 (1 in 6 predictions are perfect). The higher validation loss for ECG-PT was likely due to the high proportion

---

[1]The BIDMC data set is made available under the ODC Attribution license, and is available at https://physionet.org/content/bidmc/1.0.0/

of abnormal heart rhythms in the training and validation datasets. The full training loss curves are shown in Figure 7.

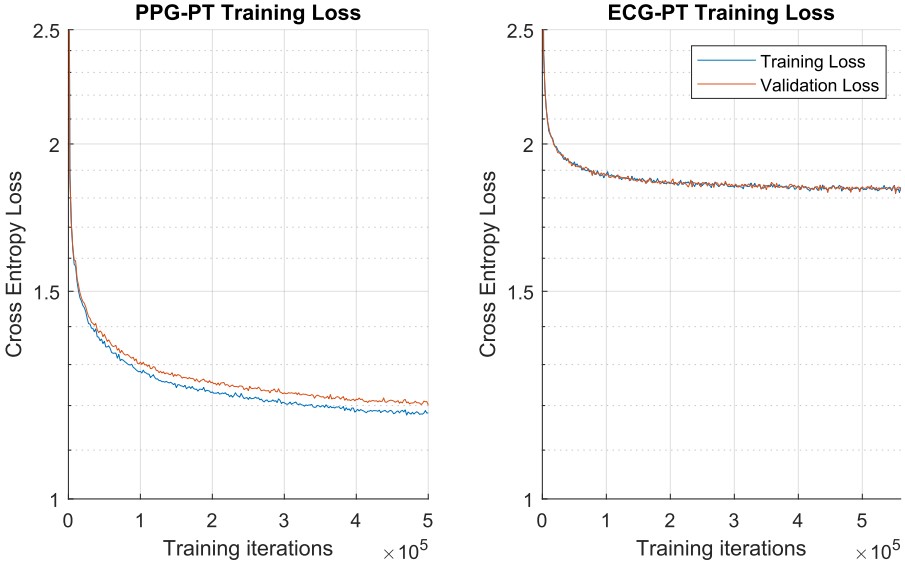

Figure 7: Training and validation loss curves for PPGPT and ECGPT training.

### A.3.1 PRE-TRAINED GENERATIVE MODEL EVALUATION DATASET

For the evaluation of generative capabilities of each model, the pre-processed subset of the Bed-based ballistocardiography dataset (Carlson et al., 2020) (Carlson et al., 2021) was used. This dataset consists of simultaneously recorded PPG and ECG from 40 subjects, sampled at 1kHz. In the preprocessed subset, the PPG was low-pass filtered with a cut off of 10Hz and the ECG was band-pass filtered between 1 and 40Hz. As with the previous datasets, we resampled the PPG to 50Hz, and the ECG to 100Hz, before converting the signals to tokens. The 750-sample windows were extracted and split into 500 samples to form the context and 250 samples to test the prediction accuracy of the model.

### A.4 CROSS ENTROPY LOSS AND NEXT TOKEN PREDICTION VS MEAN SQUARED ERROR

Conventionally, time series transformer models produce a continuous output values for each token prediction and are thus trained with a mean squared error (MSE) loss function (Das et al., 2024) or a combination of likelihood and MSE (Liu et al., 2022). However, unlike many time series prediction tasks, we were able to leverage the highly periodic nature of the PPG and ECG signals to create models with a well-defined small vocabulary. This allowed us to train the model exclusively with a cross-entropy loss function, in a similar fashion to a conventional large language model. It should be noted that we did also experiment with continuous output values and mean squared error as a loss function, but these results were less able to capture long range trends and importantly lacked interpretability in the attention weights. A full investigation into this finding is beyond the scope of this work.

Through training for next token prediction in combination with cross entropy loss, for each part of the context window, we force tokens to update in a way which is still relevant to that specific token. For example, tokens around peaks update to become more "peak-like" as they propagate through the transformer blocks, and the same principle is the case for all other positions in each cardiac cycle. In that sense, the final layer of attention operates on a ECG or PPG signal where each token is rich in period-specific information, and attention can thus be fine-tuned in a way that is more easily interpretable to a clinician.

## A.5 Fine-tuning Methods

For fine-tuning classification tasks, AUC, F1-score, sensitivity, specificity, false positive rate and false negative rate are provided. Each cross-validation was performed 5 times in order to calculate a mean and standard deviation for these metrics.

### A.5.1 Atrial Fibrillation Detection in ECG and PPG

During the fine-tuning, the final linear layer, which converts the output from $d_{model}$ dimensions to the dimensions of the vocabulary size, was replaced with a linear layer which instead converts the output from $d_{model}$ dimensions to 1 dimension. This new output was then passed through a sigmoid activation function to scale the values between 0 and 1; the final value was then used to classify the signal as either healthy (0) or atrial fibrillation (1). A binary cross-entropy loss criterion was then used to map the output value to the true class. This same conversion of the final linear layer of the model could be used for any suitable classification or regression task, by simply scaling the number of output dimensions to be in line with the specific task. During training, just the new linear layer and the final transformer block were trained, whilst all other layers were frozen. In this example, the learning rate was maintained at $3 \times 10^{-4}$ within the AdamW optimiser.

To train and evaluate the fine-tuning of the model for classification of atrial fibrillation (AF), the "MIMIC PERform AF" dataset (Charlton et al., 2022) was used, which is a subset of MIMIC III dataset (Johnson et al., 2016). This dataset contains 20 minutes of continuous PPG and ECG recordings from 35 subjects, of whom 19 had AF and 16 did not have AF. These signals, originally recorded at a sample frequency of 125Hz, were band-pass filtered to between 1 and 15Hz and then downsampled to 50Hz in the case of the PPG, and band-pass filtered to between 1 and 45Hz and downsampled to 100Hz in the case of the ECG. A window length of 500 samples was maintained, corresponding to 10 seconds of PPG (50Hz) and 5 seconds of ECG (100Hz), and sliding windows were extracted from the data and tokenised with a shift of 50 samples each time, in order to artificially increase the volume of data for the model training. In both the cases of ECG and PPG, the models were fine-tuned in a leave-one-subject-out fashion by training on 34 subjects and testing on 1, over 1,000 iterations and with a batch size of 128. Each model took 11 minutes to fine tune on an RTX A2000 GPU, which is a fraction of the over 5 days that it took to train each of the base models.

### A.5.2 Beat Detection in PPG

As was the case with fine-tuning to detect atrial fibrillation, the final linear layer was again replaced with a linear layer which converted the output from $d_{model}$ dimensions to 1 dimension. However, instead of using just the final value in the context as was the case with AF, all values from the context were used. This 1-dimensional array was passed through a Sigmoid activation function, to map to values of 0 (no peak) or 1 (peak) for all tokens in the context. Only the new linear layer and the final transformer layer were trained. The learning rate was again maintained at $3 \times 10^{-4}$ within the AdamW optimiser, with a binary cross entropy loss function. The model was fine-tuned over 5000 iterations, with a batch size of 128.

To train and evaluate the fine-tuning of the model for beat detection, the "MIMIC PERform" training and test datasets were used (Charlton et al., 2022). Both datasets consist of real-world clinical PPG and ECG recordings from 200 subjects at a sample frequency of 125Hz, for 10 minutes per subject. The training dataset was used to train and validate, and the unseen test dataset was then used to evaluate performance of the trained model. The PPG signals were band-pass filtered between 1 and 30 Hz to remove low frequency variations, and ECG between 4 and 60Hz to isolate the higher frequency QRS complex and remove P and T waves. Labels for peaks were determined by using the inbuilt MATLAB `findpeaks` function on the filtered reference ECG data. Peaks, and the points immediately neighbouring the peaks, were labelled as 1. All other points were labeled as 0. This signal was then aligned with the filtered PPG signal to the point of maximal cross correlation, to ensure peak labels that mapped to the PPG peaks. Both the reference signal and the PPG signal were resampled to 50Hz, and the PPG signal was then tokenised as per previous examples. In the reference signal, the difference in the time between each peak should be minimal. When the standard deviation of this difference was greater than 0.3 seconds, a sign that the reference signal was of poor quality, the subject was excluded from the training data.

When testing, the criterion outlined by Charlton *et al.* (Charlton et al., 2022) was followed to allow for comparability of results. Poor quality labels were excluded from the test set by removing reference segments which had no beats for 2 seconds or more. Input segments with constant values for more than 0.2 seconds, an indication of clipping and thus data loss, were also excluded from analysis. Output peaks were determined by applying `findpeaks` to the output of the model, with a minimum height of 0.015, a maximum peak width of 0.4 seconds, and a minimum distance between peaks of 0.28 seconds (corresponding to a heart rate of 214 beats per minute). If a predicted beat was within 0.15 seconds of a reference beat, it was deemed to be correct. Results were then analysed in terms of the median F1 score (%) across test subjects.

It should be noted that the fine-tuned model produced continuous values between 0 and 1, for each position in the context, which are a measure of its confidence that a beat is present. In our analysis, we also investigated the utility of using these beat confidence values as a measure of signal quality.

### A.5.3 Detection of Premature Ventricular Contractions in ECG

For this task, we extracted a subset of over 200 subjects with premature ventricular contractions from the Chapman 12-lead electrocardiogram database (Zheng et al., 2020). Given that the length of these segments are 10 seconds, we manually cropped segments to 5 second windows PVCs occured. These were matched with over 5000 subjects with no PVCs. The ECG was extracted only from lead-I, and downsampled from 500Hz to 100Hz in accordance with the inputs to our pretrained model. Segments were again rescaled to between 0 and 100, and rounded to integers to form a vocabulary of 101 tokens. Multiple staggered windows separated by 1 token were extracted for each PVC subject, in order to match the proportion of none-PVC data. The linear layer was changed in exactly the same fashion as for the classification of atrial fibrillation, by reducing the model dimension to 1 output value, and the model was fine-tuned for 1500 iterations. Training and testing was performed with 4-fold cross validation.

### A.5.4 Four class arrhythmia detection from single-lead ECG

In this example, all 10,000 subjects of the Chapman 12-lead electrocardiogram database (Zheng et al., 2020) were used. The data was merged into 4-classes following the same protocol as (Zheng et al., 2020) and (Kiyasseh et al., 2021). Only first half of all recordings were selected, to comply with the maximum 5 second context length of our model. The resampling and tokenisation was the same as for the detection of PVCs (appendix A.5.3). Training and testing was performed with 5-fold cross validation, and repeated 5 times with different random seeds. Each model was trained for 1500 iterations.

### A.6 Generative Pre-trained Model Error Overview

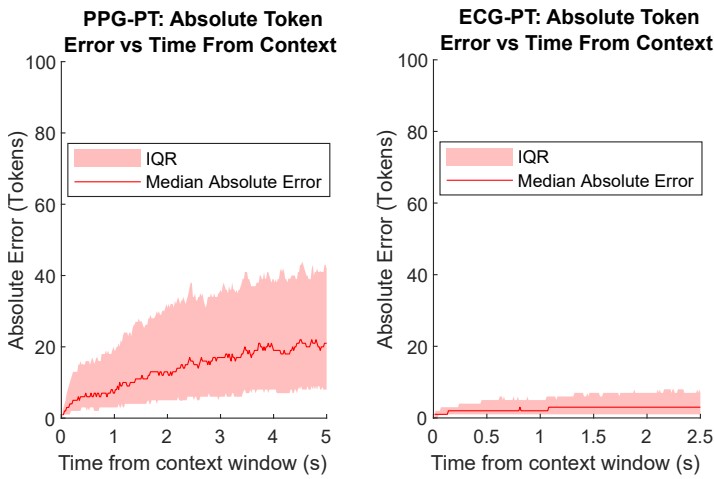

Figure 8: The median prediction errors (red solid lines) and interquartile ranges (red shaded areas) for PPG-PT and ECG-PT across all windows extracted from 40 unseen subjects in the bed-based balistocardiography dataset.

## A.7 ATTENTION DISTANCE TABLE

| Transformer Layer | Mean Layer Attention Distance for Different Models (in seconds) | |
| :---: | :---: | :---: |
| | **PPG-PT** | **ECG-PT** |
| 1 | $0.43 \pm 0.07$ | $0.33 \pm 0.11$ |
| 2 | $0.55 \pm 0.33$ | $0.99 \pm 0.26$ |
| 3 | $1.04 \pm 0.29$ | $1.17 \pm 0.29$ |
| 4 | $0.92 \pm 0.37$ | $1.11 \pm 0.10$ |
| 5 | $0.65 \pm 0.37$ | $0.58 \pm 0.11$ |
| 6 | $0.92 \pm 0.55$ | $0.51 \pm 0.22$ |
| 7 | $2.62 \pm 0.47$ | $1.79 \pm 0.26$ |
| 8 | $2.31 \pm 0.42$ | $1.88 \pm 0.23$ |

Table 1: The average look-back distance for the attention, in seconds, for all the layers (1 to 8) in the PPG-PT and ECG-PT models.

## A.8 COMPARISON WITH PRIOR WORKS ON DETECTION OF ATRIAL FIBRILLATION

For automatic detection atrial fibrillation, our result of an AUC of 0.99 over 5-second windows is comparable to the state of the art using 2-minute segments (Bashar et al., 2019). In contrast to our fine-tuned ECGPT model, this study did not provide meaningful interpretability of classifications.

For automatic detection of atrial fibrillation from PPG, it is difficult to find comparable studies on finger PPG. The majority of studies use a fully shuffled train-test pool, allowing models to learn distinct PPG morphology and noise profile of each subject and thus achieving higher accuracies for the detection of atrial fibrillation. A comparable study which did test on unseen subjects used wrist PPG in combination with the wavelet transform and a convolutional neural network. When excluding noise and taking the median of features across all 30-second windows, this study achieved an area under the curve of 0.95 (Shashikumar et al., 2017). This result is comparable to our AUC of 0.93 on 10-second windows without the exclusion of noisy subjects and 0.96 with the exclusion of noisy subjects. Crucially, this study did not present a model that was interpretable to the end user. Additionally, previous work by Torres-Soto & Ashley (2020) has demonstrated that classification performance can increase dramatically when signal quality is classified simultaneously with AF,

achieving an AUC of 0.98. A full evaluation of our results in the presence of noise is provided in Appendix A.12.

## A.9 RESULTS OF FINE-TUNING FOR BEAT DETECTION AND SIGNAL QUALITY ESTIMATION IN PPG

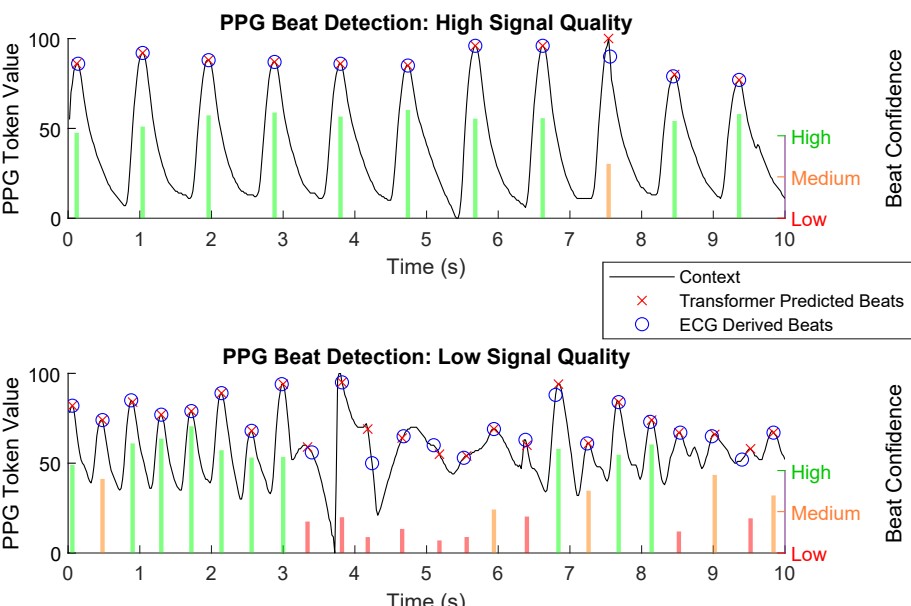

Figure 9: Illustration of fine-tuning PPG-PT for beat detection. The input context PPG signal is plotted in black, predicted beats are highlighted with red crosses and ECG derived beats (ground truth) are designated with blue circles. The beat confidence is overlaid as vertical lines, with green indicating high confidence, orange indicating medium, and red indicating low. Both an example of high signal quality (top) and an example of low signal quality (bottom) are shown.

Evaluation of the fine-tuned PPG-PT model for beat detection on the "MIMIC perform testing dataset" yielded a median F1 score of 97.7% across subjects, which is comparable to the F1 scores of qppg (Vest et al., 2018) (96.9%) and MSPTD (Bishop & Ercole, 2018) (97.5%) as found by Charlton *et al.* (Charlton et al., 2022). It was found during training that whilst the model was able to accurately distinguish beats in examples with high signal quality after just 50 to 100 iterations, training it for longer was necessary to detect beats in examples with visibly poor signal quality. In addition to predicting the location of beats, our fine-tuned PPG-PT model also provides a continuous value between 0 and 1 at each location. This can be thought of as a "quasi-confidence estimate" in the prediction that a beat is indeed present at that location. Given that test dataset is clinical data, and thus the majority of the data is high quality, values in the bottom 10% of confidence were labeled as low confidence, values between 10% and 25% were labeled as medium confidence, and values in the top 75% were labeled as high confidence.

Observe in Figure 9 that in both the cases of high signal quality and low signal quality, our fine-tuned beat detector was able to correctly identify all PPG beats, within 150ms of the ECG derived ground truth beats. Importantly, the confidence estimates also aligned with signal quality, with lower confidence in areas of poor signal quality. The confidence estimates also carry over to the previous AF dataset. Beat detection was applied independently to "MIMIC Perform AF" and it was found that two subjects with the lowest classification accuracy, and visibly poor PPG signal quality, also had the lowest mean beat confidence. Conversely, the subject with the highest mean classification accuracy had the highest signal quality, as indicated by the beat confidence.

It should be emphasised that the PPG-PT model was pretrained on finger-based clinical data, and has been fine-tuned on clinical data. Wearable sensor data, such as those from the wrist (Rajala et al.,

2018) or the ear (Davies et al., 2020) have different morphology to that of the finger. It is likely that further training would be necessary for these models to be transferable to wearable PPG data.

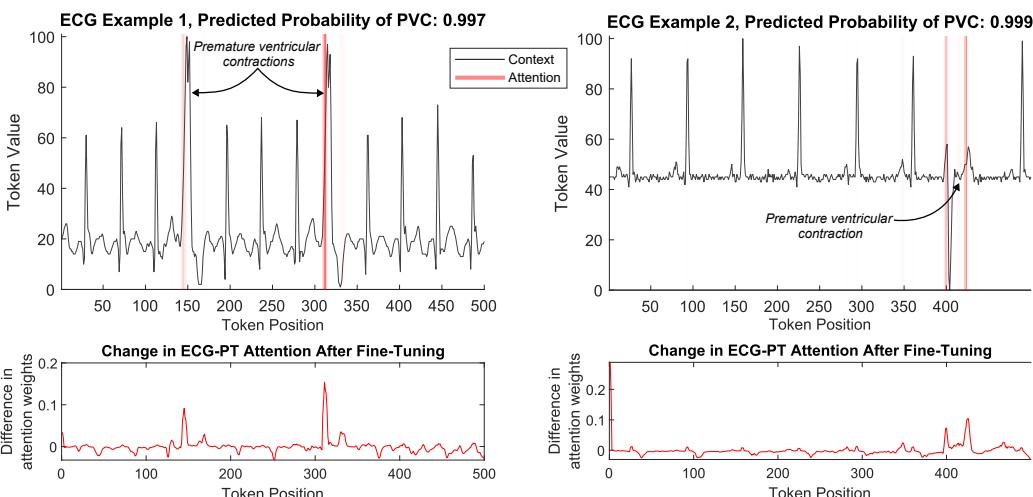

Figure 10: Changes in final layer attention weights of the base ECG-PT models when fine-tuned to classify premature ventricular contractions. The model input context is shown in black, with attention overlaid with red bars of varying transparency based on the change in attention weights. The full difference in attention weights is shown below each plot in red. The attention weights of the fine-tuned ECG-PT model clearly shift to the premature ventricular contractions.

## A.10   FINE-TUNING TO DETECT PREMATURE VENTRICULAR CONTRACTIONS IN ECG

In order to further demonstrate the capabilities of our model for interpretable classification, we fine-tuned our model our model to detect premature ventricular contractions (PVCs) in ECG. Unlike AF, the premature ventricular contraction is also characterised by a clear morphological change in the ECG (Yamada, 2019). Given this, classification of PVCs were selected for further interpretability analysis in addition to AF, given that the presence of a PVC is clear for clinicians and non-clinicians alike. A subset of over 200 subjects with PVCs was created from the Chapman dataset (Zheng et al., 2020), and matched with over 5000 subjects without PVCs. Shifted windows were extracted from subjects with PVCs in order to balance the proportion of PVC to non-PVC dataset. To evaluate our model, we performed 4-fold cross validation, ensuring no subject leakage between train and test, and repeated with 5 random seeds. The resulting AUC of 0.99 is reported in Table 2 along with further performance metrics. The interpretability analysis, which looks at the shift in final layer attention weights from the pretrained model to the fine-tuned model, is shown in two examples in Figure 10. It can be seen that attention shifts to regions of PVCs, in order to inform the classification of a PVC.

## A.11   SUMMARY OF FINE-TUNING RESULTS

The 4-class arrhythmia detection used only lead-I ECG to classify four different arrythmias in the Chapman dataset (Zheng et al., 2020), and achieves a multi-class AUC of 0.97, which compares to an AUC of 0.90 by CLOCS (Kiyasseh et al., 2021) on the same data. Other accuracies and comparisons to the literature are included in their relevant sections in the main body and appendix.

| Performance Metrics for Different Tasks (Mean ± Standard Deviation)* | | | | | | |
|---|---|---|---|---|---|
| Task | AUC | F1 Score | Sensitivity | Specificity | FPR | FNR |
| AFib (ECG) | 0.991 ± 0.002 | 0.964 ± 0.005 | 0.978 ± 0.006 | 0.940 ± 0.017 | 0.060 ± 0.017 | 0.022 ± 0.006 |
| AFib (PPG) | 0.929 ± 0.003 | 0.873 ± 0.002 | 0.947 ± 0.005 | 0.736 ± 0.007 | 0.264 ± 0.007 | 0.053 ± 0.005 |
| AFib (PPG - clean) | 0.976 ± 0.001 | 0.932 ± 0.001 | 0.952 ± 0.004 | 0.880 ± 0.004 | 0.120 ± 0.004 | 0.048 ± 0.004 |
| Beat (PPG) | - | 0.977 ± 0.001 | 0.989 ± 0.000 | - | 0.028 ± 0.001 | 0.017 ± 0.002 |
| PVC (ECG) | 0.987 ± 0.002 | 0.951 ± 0.003 | 0.940 ± 0.005 | 0.960 ± 0.003 | 0.040 ± 0.003 | 0.060 ± 0.005 |
| 4-class arrhythmia (ECG) | 0.974 ± 0.001 | 0.889 ± 0.001 | 0.884 ± 0.002 | 0.967 ± 0.000 | 0.033 ± 0.000 | 0.116 ± 0.002 |

Table 2: Performance metrics for various fine-tuning tasks on ECG and PPG data. *for beat detection these metrics were calculated on top of the median across subjects for a given repeat, as is the convention in the PPG beat detection literature.

## A.12 THE EFFECTS OF NOISE ON CLASSIFICATION

The presence of noise is a principal issue in the analysis of physiological time-series data (Charlton et al., 2023). Previous work on classification of in PPG AF has demonstrated that classification performance can increase dramatically when signal quality is classified simultaneously with AF (Torres-Soto & Ashley, 2020). To this end, we investigated the effects of noise on our classification of AF, by employing our fine-tuned beat detector (Appendix A.9) for signal quality estimation in our AF dataset. It is noted in Figure 11(a) that the majority of falsely classified test segments have low signal quality. By removing the bottom 15% of signal qualities from the test set, as determined by average beat confidence from our beat detector, the false positive rate is reduced from 26% to 12%, and the AUC is increased from 0.93 to 0.98. In Figure 11(b), an example workflow is shown, in which only signals that have high signal quality and are predicted as AF are flagged for a clinician.

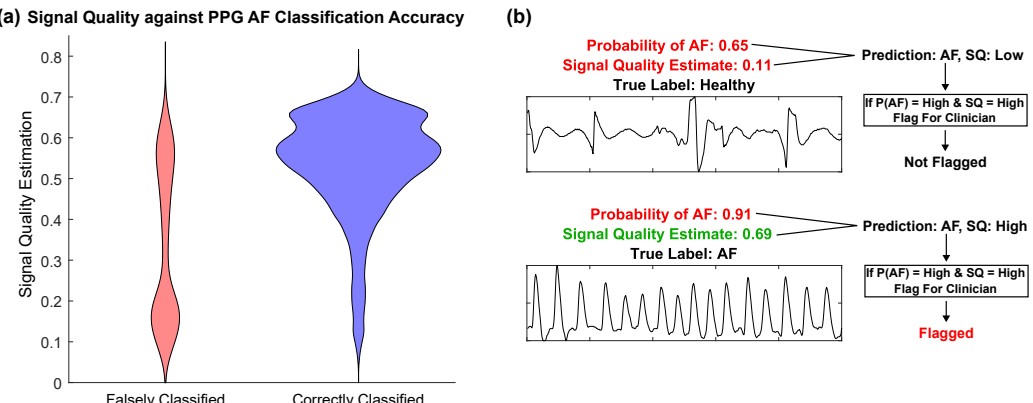

Figure 11: The effects of signal quality on the classification of atrial fibrillation in PPG. Signal quality is determined by calculating the average beat confidence, as outputted from our fine-tuned PPG beat detector. (a) Violin plots showing the distribution of signal quality for test segments that were falsely classified, and test segments that were correctly classified. (b) An example workflow which demonstrates how signal quality can be incorporated to reduce the proportion of false positives that are flagged to a clinician.

## A.13 FUTURE WORK

Many tasks require inputs of longer context lengths, such as sleep staging in which 30 seconds is the normal interval but for measurements of the autonomic nervous system contexts of 5 minutes may be required (Li et al., 2021). Future work should aim to scale up the model to larger context lengths, whilst trying to maintain efficiency, by employing methods such as Sparse Attention (Roy et al., 2021). Moreover, the models should be scaled up to account for applications that require novel sensor locations, such as in-ear PPG (Davies et al., 2022) and ear-ECG (von Rosenberg et al., 2017).

## A.14 ATTENTION METHODS

Furthermore, due to the inherent periodicity of heart time-series data, we can isolate the responses of different attention heads by averaging attention maps over time. This method is explained in more detail in Figure 12.

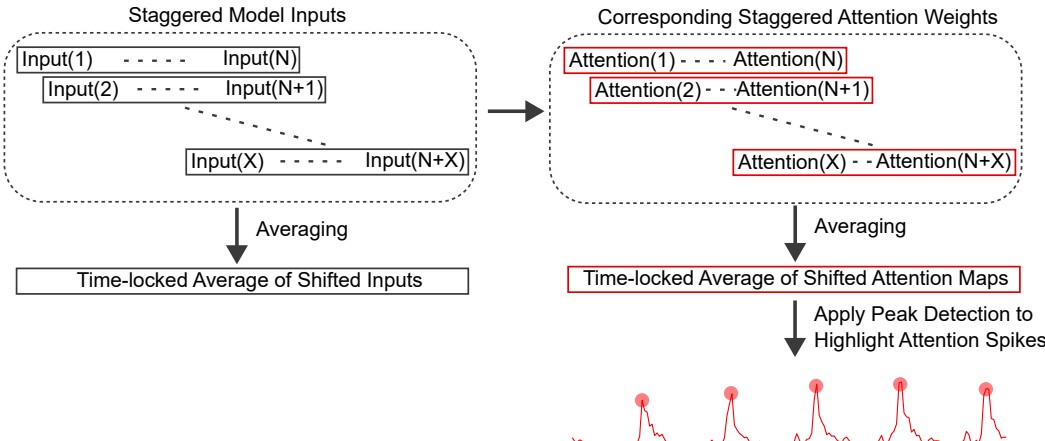

Figure 12: An explanation of the time-shifted attention algorithm that was used to detect which features different self attention heads spiked on, for PPG and ECG.

