# OpenReview forum: "Interpretable Pre-Trained Transformers for Heart Time-Series Data"
_ICLR.cc/2025/Conference — Submitted to ICLR 2025_

### Official Review · Reviewer_FHB9 · 2024-10-27

**Soundness:** 3
**Presentation:** 4
**Contribution:** 3
**Rating:** 8
**Confidence:** 4

**Summary:**

The paper aims to use generative pre-trained transformer (GPT) framework to predict next token for ECG and PPG signals with a focus in interpreting the decision-making of these large language models. This was done through observing the behavior of the attention heads in different transformer layers and aggregating them. The experiments reveal that the next token generation looks cycles in the nearest past more than that of the farthest past in the context. Qualitative interpretation reveals the ECG and PPG GPT models look into important signal components. These GPT models were fine tuned for a downstream task of atrial fibrillation classification with interpretation highlighting irregular occurrence of a beat w.r.t. the previous beat which is interesting. Overall, the problem formulation is logical, methods are well designed and explained, and the discussion clearly articulates the explainability the GPT models can offer.

**Strengths:**

- Related GPT based interpretation studies for ECG and PPG exists in literature, but the experimentation carried out in this study is rigour and systematic in revealing the explanations for firstly, the generation of next token and finally, extend the idea for a downstream classification task.

- The idea of decoding attention head for explanations are common in the CV and NLP domain, but extending it to physiological time-series can be seen a contribution.

- Data split follows a subject-wise separation to avoid data leak in training and testing.

**Weaknesses:**

Major:
- The explainability method of GPT models was shown to focus on previous cycles meaning that it can observe a beat w.r.t the previous one which is why the downstream task interprets well where distance between two consecutive beat is important criteria. However, this might not be the case for other common tasks such as sleep staging where the input in 30 second ECG or PPG signal and separating sleep stages such as wake, light sleep, deep sleep and REM sleep manifests from HR variability. Another experimentation for a downstream task would be interesting to see if the proposed explainability idea fits well to provide clinical domain specific interpretation, given the fact that GPT models were found to be useful in these downstream tasks.

- Few related GPT interpretability studies were referenced. More references should be included to have a broad picture based on GPT interpretablity. For example,
a. Creswell, A., Shanahan, M., & Higgins, I. (2022). Selection-inference: Exploiting large language models for interpretable logical reasoning. arXiv preprint arXiv:2205.09712.
b. Ben Melech Stan, G., Aflalo, E., Rohekar, R. Y., Bhiwandiwalla, A., Tseng, S. Y., Olson, M. L., ... & Lal, V. (2024). LVLM-Intrepret: an interpretability tool for large vision-language models. In Proceedings of the IEEE/CVF Conference on Computer Vision and Pattern Recognition (pp. 8182-8187).

Minor:
- Figure 3 captions needs to be corrected a-d.

**Questions:**

- 10 second segments were extracted from recordings from where tokens were generated, are these tokens represent whole segment data or beats were isolated to form tokens?

- The prediction or generation of tokens from GPT models is unclear. Experiment says a single token is generated for a given context length, then it needs to be clearly mentioned how the generation process continued to generate to produce, say Figure 1, of 15 second PPG or ECG data which was compared against the original signal?

---

> ### Author Response · Authors · 2024-11-21
> **Response to Reviewer FHB9 [Part 1]**
>
> We thank the reviewer for highlighting the interpretability of our network as a strength, and specifically down to the level of decoding of individual attention heads that respond to physiologically relevant features in PPG and ECG. We have responded to the reviewer’s comments below and believe that they have improved the quality of our manuscript.
>
> >•	The explainability method of GPT models was shown to focus on previous cycles meaning that it can observe a beat w.r.t the previous one which is why the downstream task interprets well where distance between two consecutive beat is important criteria. However, this might not be the case for other common tasks such as sleep staging where the input in 30 second ECG or PPG signal and separating sleep stages such as wake, light sleep, deep sleep and REM sleep manifests from HR variability. Another experimentation for a downstream task would be interesting to see if the proposed explainability idea fits well to provide clinical domain specific interpretation, given the fact that GPT models were found to be useful in these downstream tasks.
>
> Thank you for this suggestion. In our experience with HRV, we have found that generally inputs require time greater than 2 minutes, and usually 5 minutes, to account for the low frequency variations between 0.04 Hz (VLF band in HRV) and 0.1 Hz (LF band in HRV). Even at 30 seconds, this would require a 3-fold increase in context length for our current PPG model, and a 6-fold increase for our current ECG model. To maintain efficiency, implementing such a task would likely require the use of algorithms such as sparse attention (Roy et al, 2021) [1]. We have added this to a future work section, in Appendix A.13.
>
> For a clinical domain specific task, and to demonstrate that our model’s interpretable attention mechanism can focus on morphological features as well as timing-based features, we have added an extra downstream task for the classification of premature ventricular contractions. Premature ventricular contractions manifest in clearly visible morphological changes in the ECG waveform, and thus represent a clear litmus test for the interpretability of our model. As with all other tasks, we test only on unseen subjects. Our interpretability analysis (see Figure 10, Appendix A.10) reveals that, as desired, attention clearly shifts to regions where PVCs are present.
>
> > •	Few related GPT interpretability studies were referenced. More references should be included to have a broad picture based on GPT interpretablity. For example, a. Creswell, A., Shanahan, M., & Higgins, I. (2022). Selection-inference: Exploiting large language models for interpretable logical reasoning. arXiv preprint arXiv:2205.09712. b. Ben Melech Stan, G., Aflalo, E., Rohekar, R. Y., Bhiwandiwalla, A., Tseng, S. Y., Olson, M. L., ... & Lal, V. (2024). LVLM-Intrepret: an interpretability tool for large vision-language models. In Proceedings of the IEEE/CVF Conference on Computer Vision and Pattern Recognition (pp. 8182-8187).
>
> We have expanded our introduction (additions highlighted in blue), and added Appendix A.14 to clarify our attention method relative to these references.  We have made sure to specifically mention LVLM-interpret, which exhibits interpretability in images with a language prior, as this shares parallels with the interpretation of physiological time series with a “classification” prior.
>
> >•	Figure 3 captions needs to be corrected a-d.
>
> Thank you, we have corrected this figure caption.

---

> > ### Author Response · Authors · 2024-11-21
> > **Response to review FHB9 [Part 2]**
> >
> > > •	10 second segments were extracted from recordings from where tokens were generated, are these tokens represent whole segment data or beats were isolated to form tokens?
> >
> > In our work, each time series sample corresponds to an individual token. We were able to leverage the periodicity of physiological time-series to construct a finite vocabulary size, in which all possible token values lie. This is in contrast to other time-series transformer methods, which often group samples into patches before embedding them (Nie et al, 2023) [2] (Das et al, 2023) [3]. This is one of the reasons why our method leads to an interpretable network, as our attention mechanism focuses on individual time points. We have now clarified this in Section 3.1 “Tokenisation and Embedding” (highlighted in blue).
> >
> > > •	The prediction or generation of tokens from GPT models is unclear. Experiment says a single token is generated for a given context length, then it needs to be clearly mentioned how the generation process continued to generate to produce, say Figure 1, of 15 second PPG or ECG data which was compared against the original signal?
> >
> > Thank you for this point. We generate one token at a time. The generated token is then appended to the context and used to generate the next token. This process is repeated 250 times to generate the extra 5 seconds of PPG or extra 2.5 seconds of ECG for comparison purposes. We have improved our explanation of this method in Section 4 of the manuscript, with the clarification: “*The single generated token can then be appended to the previous context, which is then used to generate another token. This process can be repeated, generating one token at a time, until a maximum number of generated tokens has been reached.*”
> >
> > [1] Aurko Roy at al. Efficient Content-Based Sparse Attention with Routing Transformers. Transactions of the Association for Computational Linguistics, 9:53–68, 02 202
> >
> > [2] Yuqi Nie et al. A Time Series is Worth 64 Words: Long-term Forecasting with Transformers, arXiv 2211.14730, 2023
> >
> > [3] Abhimanyu Das et al. A decoder-only foundation model for time-series forecasting arXiv: 2310.10688, 2024

---

> > ### Comment · Reviewer_FHB9 · 2024-11-27
> >
> > The authors seem addressed the reviewer concern and updated the manuscript which looks good now.

---

### Official Review · Reviewer_85Mv · 2024-11-02

**Soundness:** 3
**Presentation:** 2
**Contribution:** 3
**Rating:** 8
**Confidence:** 5

**Summary:**

This manuscript presents a novel approach to representation learning for heart time-series data (typically PPG and ECG in this work) using generative pre-trained transformers (GPTs). The idea comes from the NLP domain where GPTs have shown remarkable capabilities in learning representations from text data, and the fact that the (pre-)training process for GPTs is very simple, only next-token prediction, also motivated this work, as claimed by the authors. The authors conducted a very detailed attention mechanism analysis to interpret the learned representations of the pre-trained GPTs.

**Strengths:**

1. To the best of my knowledge, this is the first work that applies generative pre-training to representation learning for heart time-series data, where previously contrastive learning methods were the most popular (also inherited from other domains like voice recognition). This is a novel and interesting idea.

2. The attention mechanism analysis is very detailed and provides a clear understanding of the learned representations, and how the transformer layers work on the heart time-series data.

**Weaknesses:**

1. The datasets used in the experiments are not comprehensive enough. The authors used the CinC2020 dataset, which is superseded by the CinC2021 dataset. The latter dataset is more comprehensive and contains more data. Moreover, there are other larger datasets available, such as the CODE-15% dataset (https://zenodo.org/records/4916206), etc.

2. The authors did not compare their method in their numerical experiments with other representation learning methods, such as contrastive learning-based methods (for example CLOCS (https://proceedings.mlr.press/v139/kiyasseh21a/kiyasseh21a.pdf)), to show the effectiveness of their method.

3. Not enough downstream tasks are conducted to evaluate the learned representations (or the pre-trained GPTs).

**Questions:**

See the "Weaknesses" section.

---

> ### Author Response · Authors · 2024-11-21
> **Reponse to reviewer 85Mv**
>
> We appreciate the reviewers feedback and positive comments about the strengths of our work, including agreement with the clear interpretability of our network. We also appreciate the reviewer’s suggestions for improvements and believe that they have improved the quality of our manuscript.
>
> > 1.	The datasets used in the experiments are not comprehensive enough. The authors used the CinC2020 dataset, which is superseded by the CinC2021 dataset. The latter dataset is more comprehensive and contains more data. Moreover, there are other larger datasets available, such as the CODE-15% dataset (https://zenodo.org/records/4916206), etc.
>
> Thank you for this suggestion. In our pre-trained model, we purposefully used a similar ratio between the number of training tokens and model size to what is currently used in large language model training, such as in Llama. We have referenced this in Subsection 3.2 “Architecture and training”.  For pre-training, adding more data than was available in CinC 2020 would be necessary only when scaling the model size. However, we find that for most tasks this may not be necessary as we achieve our goals of i) interpretability and ii) comparable and even SOTA performance in all downstream tasks. For fine-tuning, we recognise the importance of evaluating our model on this more comprehensive dataset to enhance result reliability. Thus, we have added two new fine-tuning tasks that utilise data that was added in the expanded CinC 2021 dataset, namely the Chapman 12-lead ECG dataset.
>
> > 2.	The authors did not compare their method in their numerical experiments with other representation learning methods, such as contrastive learning-based methods (for example CLOCS (https://proceedings.mlr.press/v139/kiyasseh21a/kiyasseh21a.pdf)), to show the effectiveness of their method.
> 3.	Not enough downstream tasks are conducted to evaluate the learned representations (or the pre-trained GPTs).
>
> We thank the reviewer for this suggestion. We have implemented two more fine-tuning tasks for ECG, taking our total number of downstream tasks to five. We have specifically added the 4-class arrythmia classification with the Chapman dataset to directly compare our performance numerically to CLOCS, which beats other contrastive learning-based methods.  We have included our performance on this task and other downstream tasks in a new summary table, in Appendix A.11, as well as a summary paragraph of comparisons to other architectures in the literature. We have also expanded our related work section to include works such as Kiyasseh et al 2021.
>
> Whilst the performance of our model is comparable with the state-of-the-art performance in methods suggested by the reviewer for all downstream tasks, the paramount point is that no works, in either our previous literature review or our now extended literature review, have convincing interpretability. This remains the key contribution of our work. To this end, we have also included further interpretability analysis for the downstream task of detecting premature ventricular contractions (PVCs), using a subset of the Chapman dataset (see Appendix A.10, and Figure 10). Whilst the interpretable attention of our AF downstream task focused exclusively on timing, the attention in the PVC task highlights the distinct changes in morphology that occur with a PVC. This observation further highlights the robustness and adeptness of our model's interpretability in handling diverse physiological signal characteristics, owing to the training method we employed.
>
> In our final submission, we will provide a public repository, where we open source all the model files and code, and even provide a suite of GUIs designed for interpretability. This is aimed at making our models easy to implement, for both AI researchers and medical professionals, and in this way is an attempt to bridge the gap between AI research in healthcare and real-world implementation.

---

> > ### Comment · Reviewer_85Mv · 2024-11-27
> >
> > The new experiments improved the credibility of this work.

---

### Official Review · Reviewer_Wrus · 2024-11-04

**Soundness:** 3
**Presentation:** 3
**Contribution:** 3
**Rating:** 6
**Confidence:** 3

**Summary:**

This paper introduces two interpretable pre-trained transformer models (PPG-PT and ECG-PT) adapted from the GPT framework for analyzing heart time-series data from PPG and ECG signals.  The authors demonstrate that these models develop interpretable attention patterns corresponding to physiologically meaningful cardiac cycle features.

**Strengths:**

- Interpretability analysis on physiological time series
- The model was tested on two types of physiological time series: PPG and single-lead ECG.

**Weaknesses:**

1. The model's novelty is questionable, as the PPG-GPT has previously been explored across multiple tasks (Chen et al., 2024). It will be helpful to specify how the current work differs from it.
2. The manuscript does not compare models on different PPG or ECG tasks compared to previous PPG-GPT (Chen et al., 2024). The comparison with one baseline per PPG and ECG is mentioned only in the appendix. Importantly, the models are compared on performance using different setups.
3. The experiments have not been performed across multiple runs, so the variability (std or IQR) of the performance is not clear. Consider 5-fold or 5 seeds.
4. Consider additional metrics, for example, sensitivity, specificity, false-positive rate, false-negative rate, and F1 score for AF.
5. The majority of the paper focuses on interpretability. However, it is difficult for the machine-learning community to validate clinical utility claims that require experience in reading ECG or PPG. Have you collaborated with clinical experts to validate the interpretability claims? Could you provide more evidence that the examples you have shown guarantee the presence of AF?

- Chen, Zhaoliang, et al. "Adapting a Generative Pretrained Transformer Achieves SOTA Performance in Assessing Diverse Physiological Functions Using Only Photoplethysmography Signals: A GPT-PPG Approach." AAAI 2024 Spring Symposium on Clinical Foundation Models. 2024.

**Questions:**

1: You have considered very clean datasets and samples; seeing the behavior across different noisy scenarios would be highly important. Since we could achieve very high performance on high-quality data with an F1 score of 0.96 with DeepBeat (Torres-Soto et al., 2020).

2: Attention maps as an interpretability tool have been successfully explored (Zhao et al., 2023). It needs to be clarified that you did something new methodologically; it seems more of an application. For example, you have details on using the `findpeaks` function in Matlab, but it is poorly structured overall. It would be great if you would summarize the methodology as a pseudocode and suggest choices.

- Zhao, Haiyan, et al. "Explainability for Large Language Models: A Survey." arXiv preprint arXiv:2309.01029 (2023).
- Torres-Soto, Jessica, and Euan A. Ashley. "Multi-task deep learning for cardiac rhythm detection in wearable devices." NPJ digital medicine 3.1 (2020): 116.

---

> ### Author Response · Authors · 2024-11-21
> **Response to Reviewer Wrus [Part 1]**
>
> We appreciate the reviewer’s feedback, including highlighting the interpretability of our work as a strength. We have responded to all of the reviewer’s comments below, and believe that the reviewers comments have positively impacted our manuscript.
>
> > 1.	The model's novelty is questionable, as the PPG-GPT has previously been explored across multiple tasks (Chen et al., 2024). It will be helpful to specify how the current work differs from it.
>
> We thank the reviewer for pointing us to this work by Chen et al. The major difference between our work and the previous PPG-GPT work is that they tokenise based on patches, as is common in time series transformers, rather than based on individual time points to form a finite vocabulary, as is the case with our work. Moreover, rather than predicting the next token like GPT, they interpolate patches bidirectionally in a similar way to BERT. Importantly, in the case of Chen et al, this leads to a network which does not offer interpretability. We anticipate this is due to i) a significantly larger model implemented combined with patch processing and ii) to the loss function used in training. Interpretability is critical for the acceptance of AI models in healthcare, and this is why we have made these specific design choices in our work in order to create interpretable models. We have updated our related work section to include this discussion (highlighted in blue).
>
> > 2.	The manuscript does not compare models on different PPG or ECG tasks compared to previous PPG-GPT (Chen et al., 2024). The comparison with one baseline per PPG and ECG is mentioned only in the appendix. Importantly, the models are compared on performance using different setups.
>
> We thank the reviewer for this suggestion. We have now extended our downstream tasks to further evaluate our models and provide more comparisons in terms of accuracy with other pre-trained models in the literature. We have focused on implementing more arrythmia based tasks, namely 1) 4-class arrythmia detection (supraventricular tachycardia, atrial fibrillation, sinus bradycardia and sinus irregularity), 2) detection of premature ventricular contractions.  These tasks were chosen due to their medical relevance and suitability for an interpretable model capable of aiding medical practitioners.  On the 4-class arrythmia detection in ECG, we achieve an AUC of 0.97 with just a single lead, that beats popular contrastive learning-based methods (e.g CLOCS, Kiyasseh et al 2021 [1], AUC 0.90). On beat detection in finger PPG, our average F1 score of 0.98 corresponds to state-of-the-art performance compared with MSPTD (Bishop et al, 2018 [2]) . On the PVC task, we extend our interpretability analysis and achieve an AUC of 0.99 (Appendix A.9, A.10, A.11).
>
> > 3.	The experiments have not been performed across multiple runs, so the variability (std or IQR) of the performance is not clear. Consider 5-fold or 5 seeds.
> 4.	Consider additional metrics, for example, sensitivity, specificity, false-positive rate, false-negative rate, and F1 score for AF.
>
> Thank you for these suggestions, we have now performed all fine-tuning tasks with 5 random seeds, to produce means and standard deviations for each metric. For all relevant tasks, we have also included all additional metrics suggested. These are summarised in a table in Appendix A.11.
>
> > 5.	The majority of the paper focuses on interpretability. However, it is difficult for the machine-learning community to validate clinical utility claims that require experience in reading ECG or PPG. Have you collaborated with clinical experts to validate the interpretability claims? Could you provide more evidence that the examples you have shown guarantee the presence of AF?
>
> To demonstrate interpretability, we purposefully chose AF, given that it is a very straightforward arrythmia to see visually. To ensure that this is communicated appropriately to machine learning researchers without medical training, we have included a thorough description of AF with citation (Wijesurendra and Casadei, 2019) [3] in Section 6, characterised as an irregular heart rate which manifests itself in rapid increases and decreases in heart rate. Our attention analysis shows that the model shifts attention specifically to regions where heart rate rapidly decreases or increases, and we have annotated this on Figure 5. Further to this, we have also extended our interpretability analysis to premature ventricular beats, which manifest clearly through morphological changes in the ECG. The attention of our model in this case shifts directly to regions with premature ventricular contractions. This extra analysis is present in Figure 10 (Appendix A.10). Note that in Figure 10 on the left there are two clear PVCs, and the fine-tuned attention mechanism spikes directly for both. On the right, there is one example of a PVC, and the attention shifts to just this one PVC.

---

> > ### Author Response · Authors · 2024-11-21
> > **Response to Reviewer Wrus [Part 2]**
> >
> > > 1: You have considered very clean datasets and samples; seeing the behavior across different noisy scenarios would be highly important. Since we could achieve very high performance on high-quality data with an F1 score of 0.96 with DeepBeat (Torres-Soto et al., 2020).
> >
> > We thank the reviewer for this comment and agree with this suggestion. We have included additional analysis based on the impact of noise in the classification of AF with PPG. We have firstly estimated the signal quality of our AF PPG data, using the independently fine-tuned beat detection model. Given these results, we have plotted the distribution of signal qualities for correct classification and incorrect classification, showing that a disproportionate number of test results that were falsely classified had poor signal quality. Further to this, we have shown that removing the bottom 15th percentile of signal quality from our results, resulted in an increase in AUC from 0.93 to 0.98, and a reduction in FPR from 26% to 12%. Importantly, we have also shown example signals of varying signal qualities, and a corresponding suggested workflow for reducing flags to a clinician based on signal quality.
> >
> > > 2: Attention maps as an interpretability tool have been successfully explored (Zhao et al., 2023). It needs to be clarified that you did something new methodologically; it seems more of an application. For example, you have details on using the findpeaks function in Matlab, but it is poorly structured overall. It would be great if you would summarize the methodology as a pseudocode and suggest choices.
> >
> > We thank the reviewer for this suggestion. Our methodology of interpreting attention differs from the application of raw attention masks, as the repurposing of our model from next token prediction to classification allows us to directly analyse the shift in attention between these tasks. Moreover, due to the inherent periodicity of heart time-series data, for isolating the feature response of different attention heads, we averaged attention maps over time. To the best of our knowledge, this has not been performed in any of the time series literature and is not something which would be applied in interpreting language models. We have added an explanation and methods figure (Appendix A.14, figure 12) to clarify this. Importantly, we have not seen such interpretability in the physiological time series literature, and this was unlocked by training the network with a finite vocabulary, next time-point prediction, and cross entropy loss. It has never been shown before that individual attention heads act on to specific morphological features of PPG and ECG, and that this full interpretability can extend to downstream tasks, such as detection of AF or premature ventricular contractions.
> >
> > We will open source our models and code, and have a created suite of GUIs to visualise this interpretability, making our models easy to implement for AI researchers and clinicians alike.
> >
> > [1] Kiyasseh et al, CLOCS: Contrastive Learning of Cardiac Signals Across Space, Time and Patients. arXiv: 2005.13249, 2021.
> >
> > [2] Steven M. Bishop and Ari Ercole. Multi-Scale Peak and Trough Detection Optimised for Periodic and Quasi-Periodic Neuroscience Data. Intracranial Pressure & Neuromonitoring XVI, pp. 189–195, 2018
> >
> > [3] Rohan S. Wijesurendra and Barbara Casadei. Mechanisms of Atrial Fibrillation. Heart, 105(24): 1860–1867, 2019.

---

> > > ### Comment · Reviewer_Wrus · 2024-11-25
> > > **Rebuttal Response**
> > >
> > > Thank you for addressing most of my concerns. I increased the score to "6: marginally above the acceptance threshold" to reflect this.

---

### Author Response · Authors · 2024-11-25
**Summary of our revisions and response to reviewers:**

We would like to thank all the reviewers for their insightful feedback and supportiveness of our work. The motivation for this manuscript is a lack of clearly interpretable AI in the domain of physiological time-series analysis, and to this end we are grateful that the reviewers all agreed that the core strength of our work is the clearly articulated interpretability of our network, with `[R-Wrus]` *“Strengths: interpretability analysis on physiological time series”*, `[R-85Mv]` *“The attention mechanism analysis is very detailed and provides a clear understanding of the learned representations”* and `[R-FHB9]` *“the experimentation carried out in this study is rigorous and systematic”*. We have addressed all suggestions and made the following changes, which we believe have improved the quality of our manuscript:

-  **Addition of more downstream tasks:** We have added more downstream tasks in the form of 4-class arrythmia detection from single lead ECG, and premature ventricular beat detection in ECG. This is in addition to the existing tasks of AF detection in ECG and PPG and beat detection in PPG. Through these downstream tasks, we have also integrated more recent datasets into our models, such as the Chapman dataset from the CinC 2021 challenge.

-  **Extended the analysis of interpretability for downstream tasks:** We have added more interpretability-related analysis in the classification of premature ventricular beats. Our fine-tuned model clearly shifts attention to premature ventricular beats when they occur (Appendix A.10, Figure 10). This task, like AF, was explicitly chosen given that the arrythmia is straightforward to see visually for clinicians and non-clinicians alike, meaning that we can be certain of the interpretability.

-  **Clarifying the contribution of our work against previous works:** We have extended our background literature review to include comparisons with GPT-PPG (Chen 2024), CLOCS (Kiyasseh 2020) and DeepBeat (Torres-Soto 2020). Importantly, none of the aforementioned works contain detailed interpretability analysis, which is the core contribution of our work and is of the utmost importance when applying AI tools in clinical settings.

-  **Improved robustness of the fine-tuning analysis:** We repeated all fine-tuning cross-validations with 5 different random seeds and extended the number of reported metrics whilst including a mean and standard deviation for each one.

- **Improved discussion about prior works on interpretability through attention:** We have added further discussion in our previous works section to include more references to interpretability in LLMs, such as Creswell 2022 and Zhao 2023, as well as prior literature on interpretability in large vision models (Ben Melech Stan 2024).

- **Highlighting our contribution to attention analysis methods:** We have added explanation for how we averaged attention weights across time to determine that different attention heads correspond to different features in ECG and PPG. This explanation is given in Figure 12, provided in Appendix A.14. We have also further detailed how our attention analysis in downstream tasks involves the shift in attention from pre-trained next token prediction to the fine-tuned task. In LLMs an analogous scenario would be to switch from next token prediction to sentiment analysis, and then to analyse the corresponding shift in attention.

- **Including detailed analysis on the effects of noise:** We have analysed the effects of signal quality on classification performance in AF (PPG), by using the average beat confidence from our independently fine-tuned beat detector as a proxy for signal quality. By excluding the bottom 15% of signal qualities, we have demonstrated that the false positive rate was halved. We have included visualisations of the distributions of signal quality against incorrect and correct classification, as well as an example workflow showing that signal quality estimates could be used in combination with classification probabilities to reduce false positives for clinicians. (Appendix A.12, Figure 11).

- **Addition of a future work section:** In this section we describe future avenues for research, such as scaling the model up for longer context lengths, to handle tasks such as automatic sleep staging involving heart rate variability. We also discuss the need to scale up the model to work with different morphologies of PPG, such as in-ear PPG, and lower quality ECG such as the wearable ear-ECG (Appendix A.13).

---

### Meta-Review · Area_Chair_gKLA · 2024-12-20

**Metareview:**

Summary: Although this paper received positive reviews and one borderline review but I do not believe it to be above the bar for acceptance at ICLR. This paper builds two transformer based models, one for ECGs and another for PPGs (signals used for non-invasive). These are trained via next token prediction. The work goes on to interpret what the model has learned by looking at the attention weights onto the space of inputs. They find that peak predictions are most sensitive to prior peaks, the cosine similarities of patterns such as rising and falling slopes are shared across time and that for each modality, various attention heads latch on to different parts of the signal and studying how the model's insights change when fine tuning to predict atrial fibrilation. Overall, this reads as a well written report that builds two transformers and does a study on how to interpret the model's predictions on two data modalities relevant for healthcare. My concern is not with the choice of topic, which I do think is interesting, but rather with the claims of the paper, the organization, the lack of engagement with other work in the literature on interpretability and finally the lack of clarity on what a practitioner in ML and healthcare should do with these results.

Limitations: First, it is unclear what the core technical contributions from the perspective of learning. If there are no new technical contributions, this should be stated up-front and the manuscript should instead highlight the exploratory nature of the work on a problem of interest. Given the simple methods (training transformers, using simple methods for interpreting attention weights) used here, if the contribution of this work is that transformers capture clinically relevant signals, then this should be verified via a panel of clinicians to demonstrate that using this form of interpretability improves trust and/or utility. Second, there is a large body of in deep learning at the intersection of ML and healthcare for cardiology that this work does not do much to engage with in terms of how similar the models are and this, in turn, hurts the generalizability of the findings. Finally, there is a large literature around the interpretability of transformer based models the relationship to which has not been well articulated or compared to (e.g. https://arxiv.org/abs/2005.00928, https://arxiv.org/abs/1908.07442, https://arxiv.org/abs/2406.00426,https://arxiv.org/abs/2212.14776). Is this the only method for interpretability of such models, a good one, or even the best one? I think its important to contend with these topics for this work to have value for a learning focused venue like ICLR.

**Additional Comments On Reviewer Discussion:**

The reviewers questioned whether the model’s approach was novel, noting similarities to PPG-GPT. The authors responded by highlighting how their tokenization strategy (individual time points instead of patches) and bidirectional interpolation methods differ from previous work. The authors also added additional datasets and metrics. Reviewers noted that the datasets were clean and wanted insight into performance under noisier datasets. The authors conducted additional analyses to examine how performance is affected by lower quality signals. The reviewers asked for clearer methodological descriptions and comparisons to other interpretability methods. To this, the authors added references to other GPT interpretability studies, such as LVLM-Interpret, and placed their work in a broader context.

---

### Decision · Program_Chairs · 2025-01-22

Reject